# Burden and Trends of Acute Viral Hepatitis in Asia from 1990 to 2019

**DOI:** 10.3390/v14061180

**Published:** 2022-05-28

**Authors:** Qiao Liu, Min Liu, Jue Liu

**Affiliations:** 1Department of Epidemiology and Biostatistics, School of Public Health, Peking University, Beijing 100191, China; 1610306236@pku.edu.cn; 2Institute for Global Health and Development, Peking University, Beijing 100871, China

**Keywords:** acute viral hepatitis, Asia, burden, trends

## Abstract

Asia has a very high burden of acute hepatitis; thus, a comprehensive study of the current burden and long-term trends of acute hepatitis in Asia is needed. We aimed to assess the current status and trends from 1990 to 2019 of acute hepatitis burden in Asia, using the data from the Global Burden of Diseases Study 2019 (GBD 2019) results. **Methods:** We used the data from the GBD 2019. Absolute death, incidence, and disability adjusted life years (DALY) number and rate of acute hepatitis in Asia were derived from the database from 1990 to 2019. Age-standardized mortality, incidence and DALY rates (ASMR, ASIR and ASDR) were used to compare populations in different regions and times. The estimated annual percentage change (EAPC) in the rates quantified the trends of the acute hepatitis burden. **Results:** From 1990 to 2019, the ASMR and ASDR of acute hepatitis decreased significantly at different paces, with the largest decrease in acute hepatitis C and the smallest in acute hepatitis E. The ASIR of acute hepatitis decreased relatively slowly, by an average of 0.06% (95% CI, 0.05–0.08%) per year in acute hepatitis A, 0.91% (0.64–1.18%) per year in acute hepatitis C and 0.26% (0.24–0.28%) per year in acute hepatitis E; while the ASIR of acute hepatitis B decreased by an average of 1.95% (1.08–2.11) per year. From 1990 to 2019, the incidence rate of acute hepatitis A increased in most age groups (from the age of 5 to 70), with the 50–55 years group having the fastest increase by an average of 1.81% (95% CI, 1.67–1.95%) per year. In 2019, Afghanistan had the highest ASMR (10.44 per 100,000) and ASDR (357.85 per 100,000) of acute hepatitis, and the highest ASIR was in Mongolia (4703.14 per 100,000). **Conclusions:** In Asia, the burden of acute viral hepatitis was at a relatively high level, compared with the other four continents. International cooperation and multifaceted and multisectoral actions are needed for Asian countries to eliminate viral hepatitis and to contribute to the global elimination of viral hepatitis.

## 1. Introduction

Hepatitis is a major public health issue leading to a range of health problems, including impaired liver function, acute liver injury, and acute liver failure, some of which can be fatal [1,2,3]. Hepatitis is caused by a variety of noninfectious agents and infectious viruses including the five main strains of the hepatitis virus, referred to as types A, B, C, D and E [1]. All the hepatitis viruses can cause acute hepatitis, yet only hepatitis virus B (HBV), C (HCV) and D frequently cause chronic hepatitis [4]. HAV and HEV are transmitted via the fecal-oral route and rarely develop into a chronic infection [1]. HBV and HCV are transmitted via sexual contact and drug injections, and are the most common cause of liver cirrhosis, liver cancer and viral hepatitis-related deaths [1]. In 2015, viral hepatitis caused 1.34 million deaths, and the number of deaths due to viral hepatitis has been increasing over time [4]. It was estimated that in 2015, there were 257 million people living with chronic HBV infection, and 71 million people living with chronic HCV infection, while 340 million new episodes of acute hepatitis were reported in 2017 [4,5]. However, the World Health Organization (WHO) reported that the early medications for the treatment of viral hepatitis B and C had limited effectiveness and were poorly tolerated and expensive [4]. This fact highlighted the importance of the progress of prevention interventions, such as blood safety, health-care injection safety, infection control, and harm reduction for people who inject drugs.

To achieve the goal of the 2030 Agenda for Sustainable Development, the WHO created the Global Health Sector Strategy on Viral Hepatitis, 2016–2021, which aimed to eliminate viral hepatitis as a major public threat by 2030, and promoted synergies between viral hepatitis and other health issues and aligns the hepatitis response with other global health and development strategies, plans and targets [6]. However, this ambitious goal faces many obstacles, particularly during the coronavirus disease (COVID-19) pandemic, as COVID-19 is likely to place substantial further pressure on government budgets and change policy priorities [7]. To accelerate the elimination of viral hepatitis, a Commission has drawn together a wide range of expertise to appraise the current global situation and to identify priorities globally, regionally, and nationally that are needed to accelerate progress, and they identified 20 heavily burdened countries that account for over 75% of the global burden of viral hepatitis, half of these 20 countries residing in Asia [8].

Previous studies had shown that Asia had a very high burden of acute hepatitis. East and South Asia suffered the most severe threat of hepatitis virus E (HEV) infection across the world [9]. The highest acute hepatitis C incidence was observed in Central Asia [10]. Bangladesh, followed by India and China, had the highest HEV incidence in 2017 in the world [9]. China and India approximately accounted for one-third of the global hepatitis C episodes in 2017 and hepatitis A episodes in 2019 [10,11]. In addition, China had the world’s largest burden of HBV infection [12]. With the large burden of acute hepatitis, Asia could be a major contributor to the global elimination of viral hepatitis by 2030. Therefore, a comprehensive study of the current burden and long-term trends of acute hepatitis in Asia is needed to help people understand the main issues in meeting the targets of eliminating viral hepatitis. In this study, we aimed to assess the current status and trends from 1990 to 2019 of acute hepatitis burden in Asia, using the data from the Global Burden of Diseases Study 2019 (GBD 2019) results [13]. Since GBD 2019 has not incorporated data on acute hepatitis D, only acute hepatitis A, B, C and D are discussed in this study.

## 2. Methods

### 2.1. Data Sources

We used the data obtained in the GBD 2019, which consisted of a systematic and scientific effort to quantify the comparative magnitude of health losses due to diseases by sex, age, and location over time [14]. From the website named “Global Health Data Exchange”, established by the GBD group [13], we extracted annual values and rates of deaths, incidence and disability adjusted life years (DALYs) of acute hepatitis A, B, C and E in Asian countries from 1990 to 2019, by sex and age. The general methodological approaches to estimate the acute hepatitis burden were described elsewhere [15]. Briefly, all available data on causes of death, incidence or DALY data were standardized and pooled into a single database used to generate cause-specific estimates by age, sex, year, and geography; then, multiple models, such as cause of death ensemble modelling, disease model-Bayesian meta-regression, comorbidity correction and so on, were used to estimate comparable data of different diseases across the world [15]. In particular, acute hepatitis mortality was modelled by using vital registration, surveillance, and verbal autopsy data from the cause of death database; anti-HAV seroprevalence data, hepatitis B surface antigen seroprevalence data, anti-HCV seroprevalence data and anti-HEV seroprevalence data from population-based studies and surveys were used to estimate the seroprevalence and seroincidence for further estimation of incidence and DALYs [15].

We reported the deaths, incidence and DALYs of the four types of acute hepatitis in Asian countries (overall and separately) from 1990 to 2019, and arranged these data into successive 5-year age intervals from 0–4 years to 80 plus years.

### 2.2. Statistical Analyses

The absolute values of deaths, incidence and DALYs represented the actual condition of acute hepatitis in Asia, and their relative change was defined as (*VALUES*_2019_*-VALUES*_1990_)/*VALUES*_1990_ × 100%, which showed the overall difference between 1990 and 2019. Age-standardized mortality rate (ASMR), age-standardized incidence rate (ASIR) and age-standardized DALY rate (ASDR) were calculated by applying the age-specific rates to a GBD world standard population, which were used to compare populations with different age structures or for the same population over time in which the age profiles change accordingly.

Estimated annual percentage change (EAPC) was widely used to quantify the rate trend over a specific interval. A regression line was fitted to the natural logarithm of the rates (y = α + βx + ε, where y = ln (rate) and x = calendar year). EAPC was calculated as 100 × (e^β−1^), with 95% confidence intervals (CIs) obtained from the linear regression model. In this study, overall EAPC was calculated by the annual ASMR, ASIR and ASDR of overall and each acute hepatitis in Asian countries (overall and separately), and EAPC in different age groups was calculated by the age-specific rates of deaths, incidence and DALYs. The term “increase” was used to describe trends when the EAPC and its lower boundary of 95% CI were both >0. In contrast, “decrease” was used when the EAPC and its upper boundary of 95% CI, were both <0. Otherwise, the term “stable” was used.

## 3. Results

### 3.1. Overall Acute Hepatitis Burden and Time Trends in Asia

From 1990 to 2019, the age-standardized incidence rate (ASIR) and the absolute incidence number of acute hepatitis E in Asia were the highest across the five continents. Despite the relatively lower ASIR of acute hepatitis A (second highest before 2013 and third highest after 2013), acute hepatitis B (the third highest) and acute hepatitis C (close to America, Europe and Oceania, and much lower than Africa) in Asia, the absolute incidence numbers of these diseases are all the highest in Asia, from 1990 to 2019 (Figure 1).

In Asia, acute hepatitis A had the highest age-standardized mortality rate (ASMR), ASIR and age-standardized DALY rate (ASDR) from 1990 to 2019, followed by acute hepatitis B, while acute hepatitis C and E had relatively low ASMR, ASIR and ASDR.

The ASMR of acute hepatitis A and B decreased sharply, from 3.14 and 1.22 in 1990 to 0.74 and 0.49 in 2019 (per 100,000), respectively.

The ASIR of acute hepatitis A had a slight decrease from 2268.09 in 1990 to 2218.07 in 2019 (per 100,000); however, the ASIR of acute hepatitis B had a significant decrease from 1909.53 in 1990 to 1089.91 in 2019 (per 100,000).

The ASDR of acute hepatitis A and B also decreased significantly from 174.25 and 56.50 in 1990, to 46.80 and 25.51 in 2019 (per 100,000). Acute hepatitis C and E also had a slight decrease in ASMR, ASIR and ASDR from 1990 to 2019 (Figure 2B).

In Asia, from 1990 to 2019, the ASMR and ASDR of acute hepatitis decreased significantly at different paces, with the largest decrease in acute hepatitis C (EAPC of ASMR = −4.73, 95% CI, −4.84–−4.63; EAPC of ASDR = −4.17, 95% CI, −4.38–−3.95) and the smallest decrease in acute hepatitis E (EAPC of ASMR = −3.13, 95% CI, −3.25–−3.00; EAPC of ASDR = −2.07, 95% CI, −2.23–−1.90). However, the ASIR of acute hepatitis decreased relatively slowly, by an average of 0.06% (95% CI, 0.05–0.08%) per year in acute hepatitis A, 0.91% (0.64–1.18%) per year in acute hepatitis C and 0.26% (0.24–0.28%) per year in acute hepatitis E; while the ASIR of acute hepatitis B decreased by an average of 1.95% (1.08–2.11) per year (Table 1).

### 3.2. Sex and Age Diversities in Acute Hepatitis Burden and Time Trends

Sex disparity changed differently in the four acute hepatitis from 1990 to 2019. The burden of acute hepatitis A did not change significantly, while the sex gap became larger in acute hepatitis B and E, and smaller in acute hepatitis C in 2019 than in 1990 (Figure 2A).

As shown in Figure 3, the incidence and DALY rate of acute hepatitis A decreased with age both in 1990 and 2019, while the mortality rates were both high in the younger and older populations. In 2019, the incidence and DALY rate of acute hepatitis A in children under the age of 5 was 9999.68 per 100,000 and 115.53 per 100,000, which were much higher than other age groups. The highest mortality rate of acute hepatitis A in 2019 was in the over 80 years group (2.22 per 100,000). In 1990 and 2019, the highest incidence rate of acute hepatitis B was in people aged 25–30 (2763.42 per 100,000) and 30–35 (1818.37 per 100,000), respectively. However, the death rate of acute hepatitis B increased with age, with the highest death rate of 4.45 per 100,000 in 1990 and 1.55 per 100,000 in 2019.

From 1990 to 2019, the mortality and DALY rates of acute hepatitis decreased in most age groups at different paces. Notably, the smallest decreases were all in the 10–15 years groups of the four acute hepatitis types. In the 10–15 years group, the EAPCs of the mortality rates were −2.09 (95% CI, −2.82–−1.36) for acute hepatitis A, −0.53 (−0.97–−0.08) for acute hepatitis B, −2.30 (−2.81–−1.79) for acute hepatitis C, and 0.35 (−0.13–0.83) for acute hepatitis E. The EAPCs of the DALY rates for these four acute hepatitis types were −1.90 (−2.57–−1.22), −0.73 (−1.18–−0.29), −2.27 (−2.77–−1.77) and 0.10 (−0.18–0.38) in the 10–15 years group, respectively. However, the incidence rate of acute hepatitis A increased in most age groups (from the age of 5 to 70), with the 50–55 years group having the fastest increase by an average of 1.81% (95% CI, 1.67–1.95%) per year. The over 80 group of acute hepatitis C and over 65 groups of acute hepatitis E also had EAPCs of incidence rate over 0. (Figure 4).

### 3.3. Diversities of Acute Hepatitis Burden in Asian Countries

In 2019, Afghanistan had the highest age-standardized mortality rate (ASMR) (10.44 per 100,000) and age-standardized DALY rate (ASDR) (357.85 per 100,000) of acute hepatitis among all the Asian countries, much higher than the second highest country, Pakistan (ASMR = 5.67 per 100,000, and ASDR = 214.85 per 100,000). The highest age-standardized incidence rate (ASIR) of acute hepatitis was in Mongolia (4703.14 per 100,000), followed by Yemen (4661.63 per 100,000) and Afghanistan (4480.38 per 100,000). Israel had the lowest ASIR (1322.17 per 100,000) and ASDR (3.81 per 100,000), and the second lowest ASMR (0.03 per 100,000) (Figure 5).

From 1990 to 2019, the burden of acute hepatitis in most Asian countries decreased at various paces. The largest decrease in acute hepatitis of ASMR and ASDR was observed in Turkey by an average of 13.64% (95% CI, 12.41–14.85%) and 10.65% (9.72–11.57%) per year, respectively, while the largest decrease in ASIR was found in China (EAPC = −1.33, 95% CI, −1.42–−1.23). However, both the ASMR and ASDR of acute hepatitis in Georgia increased from 1990 to 2019, by an average of 3.12% (1.47–4.79%) and 1.74% (0.69–2.80%) per year, respectively. Georgia was also the only country that had increased ASMR and ASDR of acute hepatitis from 1990 to 2019 across Asia. The ASIR in Japan (EAPC = 0.00, 95% CI, −0.09–0.08) and Nepal (EAPC = −0.01, 95% CI, −0.04–0.03) has been stable in the past 30 years (Figure 5).

Although most Asian countries had decreasing trends of overall acute hepatitis burden, there were increasing trends in some countries when considering certain types of acute hepatitis. For acute hepatitis A, the ASDR increased in Georgia (EAPC = 1.10, 0.62–1.59), the ASMR increased in Georgia (EAPC = 4.62, 2.71–6.57), and the ASIR increased in Thailand (EAPC = 1.35, 0.99–1.71), Japan (EAPC = 0.86, 0.68–1.03), South Korea (EAPC = 0.37, 0.09–0.65) and 10 other countries. The ASDR of acute hepatitis B increased in Georgia (EAPC = 3.23, 1.77–4.70) and Cambodia (EAPC = 1.25, 0.53–1.97), and the ASMR in these two countries also increased by an average of 4.15% (2.46–5.87%) and 1.83% (1.00–2.67%) per year, respectively. The ASIR of acute hepatitis C increased in seven countries, such as Armenia (EAPC = 0.53, 0.45–0.62), Mongolia (EAPC = 0.41, 0.24–0.57), Syria (EAPC = 0.38, 0.06–0.70) and so on. As for acute hepatitis E, the ASDR increased in Cambodia (EAPC = 1.99, 1.17–2.81) and Georgia (EAPC = 1.73, 0.84–2.63), the ASMR increased in Cambodia (EAPC = 2.94, 1.98–3.90), Oman (EAPC = 2.44, 2.07–2.81) and Georgia (EAPC = 2.37, 0.49–4.29), and the ASIR increased in Bhutan (EAPC = 1.09, 0.84–1.34), Yemen (EAPC = 0.56, 0.36–0.76) and five other countries (Table 1).

Despite the fact that the overall values of DALYs, deaths and incidence of the four acute hepatitis types were smaller in 2019 than in 1990 in Asia, these values in some countries largely increased in 2019 compared to 1990. As shown in Table 1, in Qatar and UAE, the DALYs of acute hepatitis increased by at least 133.34% (UAE, acute hepatitis C), and at most 393.91% (UAE, acute hepatitis B). The episode number of acute hepatitis expanded by at least 145.54% (UAE, acute hepatitis A), and at most 443.14% (Qatar, acute hepatitis C).

## 4. Discussion

To the best of our knowledge, this is the first comprehensive effort to summarize the status of acute viral hepatitis burden in Asia using data from the GBD 2019, assessing overall and age-specific rates of deaths, incidence, and DALYs of acute viral hepatitis, as well as their long-term trends and diversities between sex and different Asian countries. From 1990 to 2019, Asia had the largest acute viral hepatitis burden among the five continents all over the world. Among the four types of acute viral hepatitis that we discussed in this study, the burden of acute hepatitis A was larger than the other three. Despite the significant decrease in the ASMR and ASDR from 1990 to 2019, the decline range of the ASIR of acute viral hepatitis was barely satisfactory. Both the younger and the older populations were more affected by acute viral hepatitis, while many sexually active people also suffered from acute hepatitis B. Turkey had the largest decrease in ASMR and ASDR of acute viral hepatitis during the last 30 years, while the largest decrease in ASIR was observed in China. However, there were countries that needed to focus on acute viral hepatitis control, such as Georgia (with the largest increase in ASMR and ASDR) and Afghanistan (with the largest burden of acute viral hepatitis). With the large burden and complex situation of acute viral hepatitis in Asia, multifaceted and multisectoral actions are needed to reduce the deaths due to acute viral hepatitis and eliminate acute viral hepatitis globally.

Our results, which were consistent with the findings of previous studies, [9,10,11,16] showed that Asia had the largest absolute incidence number of acute viral hepatitis among the five continents across the world from 1990 to 2019, while the ASIR of HAV and HCV infection was the highest, and HBV and HEV infection was the second highest. To follow the global strategy on elimination, some Asian countries have developed national plans for eliminating viral hepatitis. Myanmar launched treatment for HCV in seven general hospitals in 2018 provided free of user charges, Thailand provided PrEP free under the Princess PrEP program, [17] and an extensive program was started in China in 2002 to prevent HBV transmission to newborn babies in all the Chinese provinces, which was known as the China GAVI Hepatitis B Immunization Project [18]. A vaccine to prevent HEV infection has been developed but was only licensed in China, although the WHO attempted to enable its use in areas of pressing need [19,20]. With the large burden of acute viral hepatitis and countries’ efforts regarding its elimination, Asia could be a major contributor to the global elimination of viral hepatitis by 2030.

In Asia, for the four types of acute viral hepatitis, significant decreases were observed in the ASDR and ASMR from 1990 to 2019; however, the ASIR only had slight decreases, especially for acute hepatitis A, C and E. These decreasing trends could partly be explained by the rapid scientific and technological advances in the last decades, which made it possible to control and even cure the infections, with a strong focus on prevention medicine through vaccination [21]. However, there remains multiple problems due to the complex characteristics, epidemic history and outcomes of these four types of acute viral hepatitis.

HAV and HEV are transmitted via the fecal-oral route and rarely develop into a chronic infection [1]. Although their infection usually causes self-limiting hepatitis, fulminant liver failure from hepatitis A and E may rapidly progress within a week, often resulting in death or an emergent liver transplant [20,22]. Therefore, early diagnose of hepatitis A- and E- associated fulminant liver failure and urgent decision on liver transplantation are of vital importance. Notably, our results also showed that the incidence rate of acute hepatitis A even increased in some age groups (from the age of 5 to 70). Thus, there remains the need for massive efforts to reduce the spread of hepatitis A in Asia to meet the goal of elimination in 2030. The spread of hepatitis A and E could be reduced by adequate supplies of safe drinking water, proper disposal of sewage within communities, and personal hygiene practices [20,22].

In contrast, HBV and HCV are transmitted via sexual contact and drug injections, and they are more likely develop into chronic infections [1]. For acute viral hepatitis, HAV caused the most mortality; however, HBV and HCV caused over 95% of the mortality from viral hepatitis (including acute and chronic) [4]. It was estimated that only 10% of the people living with hepatitis B and C knew their status, [23,24] which brought great challenges to its elimination [17]. The WHO recommends that all infants receive the hepatitis B vaccine as soon as possible after birth, preferably within 24 h, followed by two or three doses of hepatitis B vaccine at least four weeks apart to complete the vaccination series [25]. However, effective vaccination for HCV is still absent, so reducing the risk of exposure in health care settings and in higher risk populations are the main prevention strategies for hepatitis C [16].

In this study, we found that the burden of acute viral hepatitis was distributed heterogeneously in different populations. Our results showed that females accounted for a higher proportion of acute hepatitis B and E than males, and both the younger and the older populations were more affected by acute viral hepatitis, while many sexually active people also suffered from acute hepatitis B. It was reported that female patients tend to exhibit more severe cases of overt acute hepatitis B, indicated by a prolonged prothrombin time, [26] hence, the need to focus on the prevention against HBV infection for females. Moreover, mother-to-child transmission is the predominant mode of HBV infection in high-intermediate endemic areas, and 95% of neonates infected with HBV at birth are at risk of developing chronic infection and 15–40% of them are at risk of developing cirrhosis and liver cancer [27]. Therefore, prevention of mother-to-child transmission of HBV is one of the five core strategies of global HBV elimination by 2030 [6,12]. In 2017, 84% coverage of the hepatitis B vaccine was reported (third dose), yet only 43% coverage for timely birth doses of the hepatitis B vaccine, and the differences among countries remained huge [28]. Meanwhile, testing and treatment in older age groups should not be ignored.

Despite the fact that most Asian countries experienced a decrease in acute viral hepatitis burden, several countries, such as Georgia, Qatar and UAE, were still facing severe acute hepatitis burden. In Qatar, large numbers of immigrant workers from low-income countries have been drawn in over the last two decades, [29] indicating that immigrant growth might play an important role in driving the increasing burden of acute viral hepatitis [11]. Due to the rapid globalization processes and increasing high-risk populations of acute viral hepatitis, all the countries have a responsibility to globally eliminate acute viral hepatitis.

In conclusion, in Asia, the burden of acute viral hepatitis was at a relatively high level, compared with the other four continents. Although the ASMR, ASIR and ASDR of acute viral hepatitis decreased during the last 30 years in Asia, there remains multiple problems that need to be solved in order to meet the goal of elimination by 2030. The ASIR of acute viral hepatitis decreased at a relatively slow pace, while the incidence rate of acute hepatitis A had an uptrend in people aged 5 to 70 years from 1990 to 2019. Vulnerable groups, such as children under the age of 5, people aged over 70, and females, require more attention. Immediate actions are needed to reverse the uptrends of the ASMR, ASIR and ASDR of acute viral hepatitis in certain countries. International cooperation and multifaceted and multisectoral actions are needed for Asian countries to eliminate viral hepatitis and to contribute to the global elimination of viral hepatitis.

## Figures and Tables

**Figure 1 viruses-14-01180-f001:**
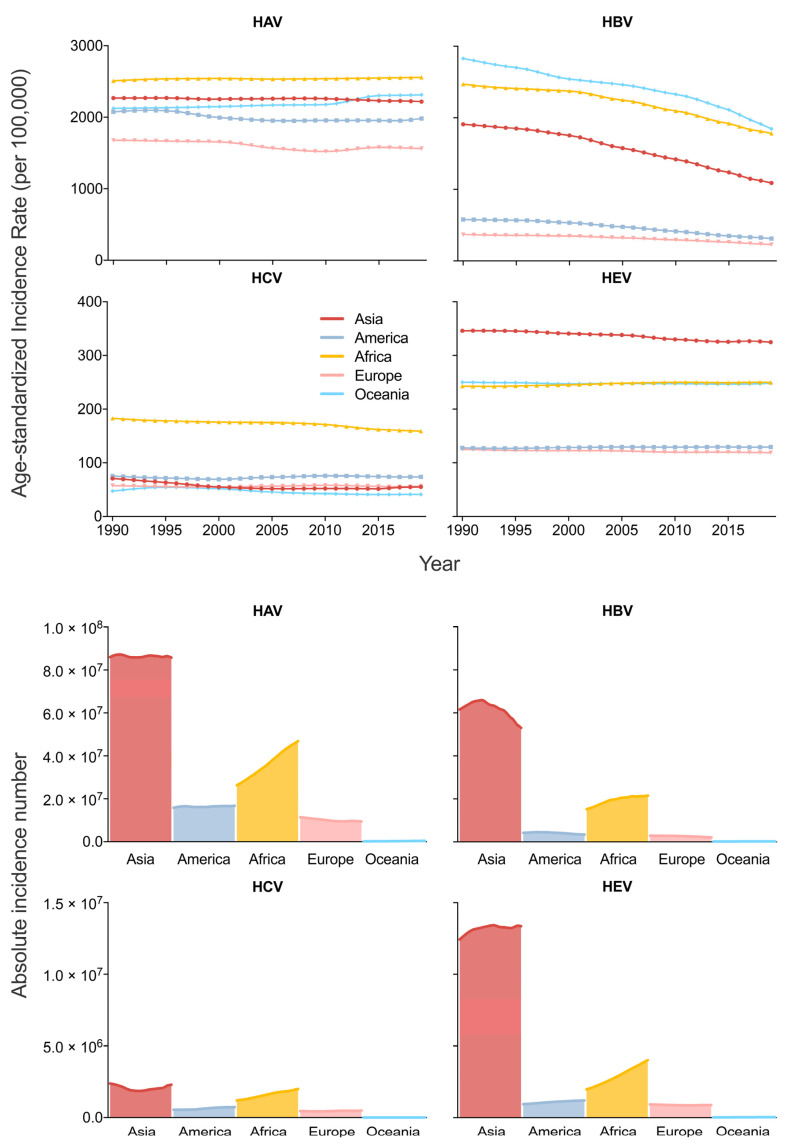
Trends of age-standardized incidence rate and absolute incidence number of acute hepatitis in Asia, America, Africa, Europe and Oceania from 1990 to 2019.

**Figure 2 viruses-14-01180-f002:**
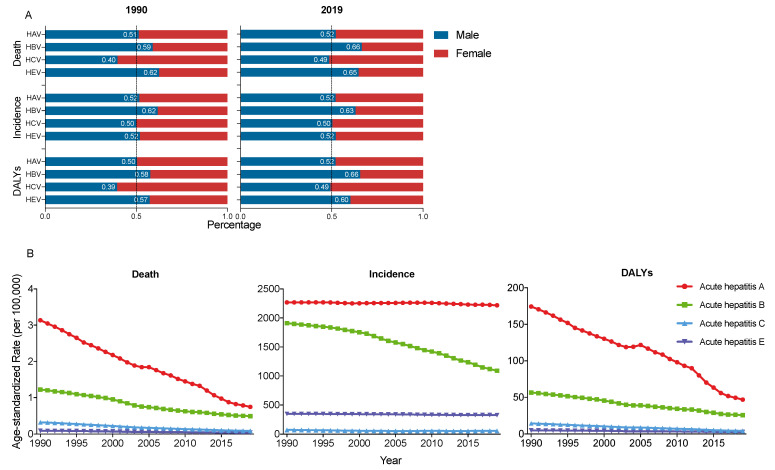
(**A**) Sex diversities in acute hepatitis in Asia in 1990 and 2019; (**B**) trends of age-standardized rate of deaths, incidence and DALYs of acute hepatitis in Asia, from 1990 to 2019. DALY, disability adjusted life year.

**Figure 3 viruses-14-01180-f003:**
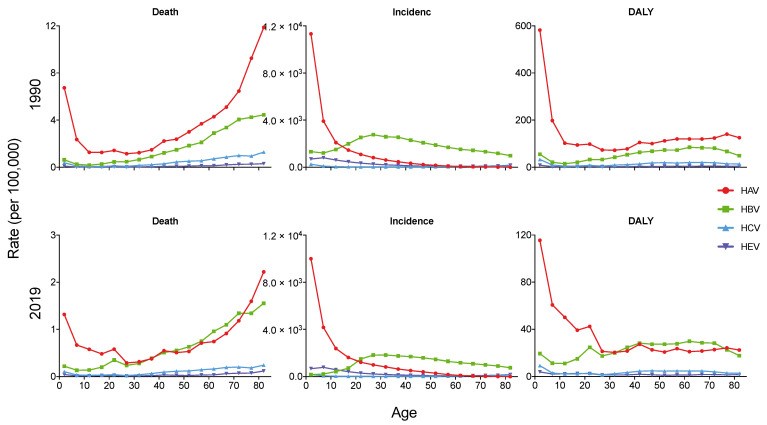
Age-specific mortality, incidence and DALY rates of acute hepatitis in Asia in 1990 and 2019. DALY, disability adjusted life year.

**Figure 4 viruses-14-01180-f004:**
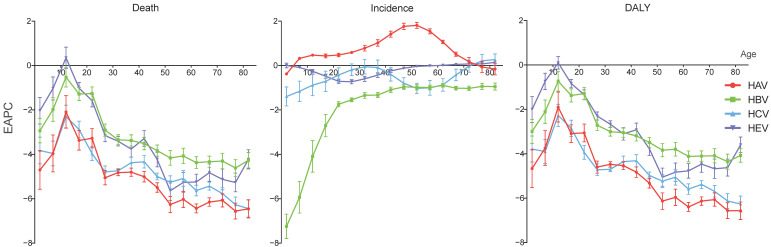
EAPCs in different age groups of acute hepatitis in Asia and the corresponding 95% CIs, from 1990 to 2019. EAPC, estimated annual percentage change; CI, confident interval.

**Figure 5 viruses-14-01180-f005:**
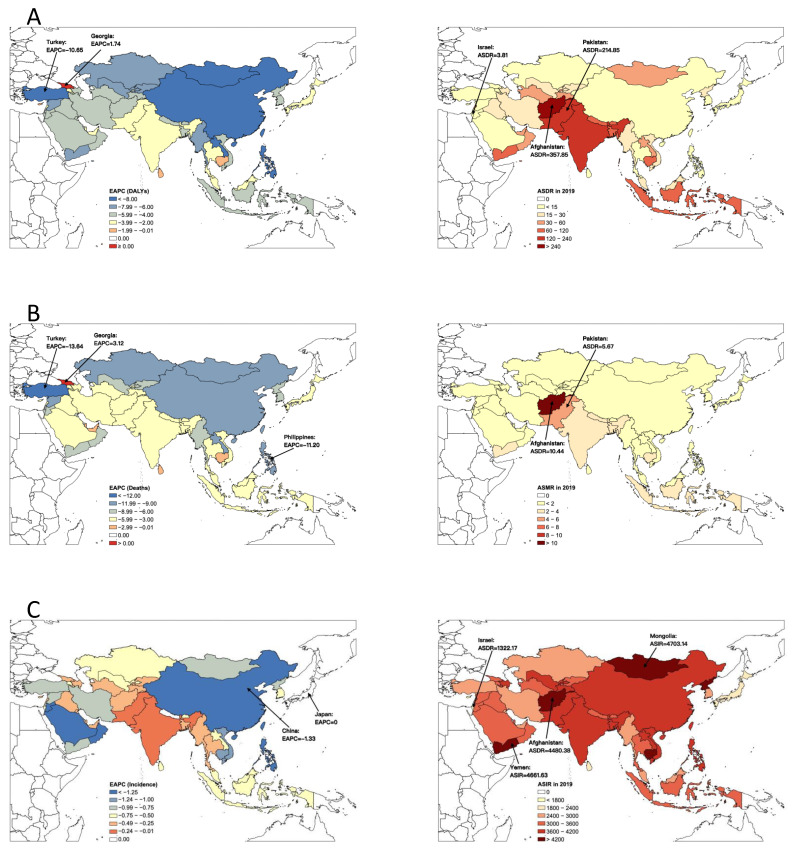
EAPCs and age-standardized rate of (**A**) DALYs, (**B**) deaths and (**C**) incidence of acute hepatitis burden in Asian countries. EAPC, estimated annual percentage change; DALY, disability adjusted life year.

**Table 1 viruses-14-01180-t001:** EAPCs and changes between 1990 and 2019 of acute hepatitis burden of overall and top five countries in Asia.

	EAPC				Changes between 1990 and 2019 (Absolute Values in 2019)			
	Acute Hepatitis A	Acute Hepatitis B	Acute Hepatitis C	Acute Hepatitis E	Acute Hepatitis A	Acute Hepatitis B	Acute Hepatitis C	Acute Hepatitis E
*DALYs*								
Asia	−4.10 (−4.59, −3.60)	−2.78 (−2.88, −2.67)	−4.17 (−4.38, −3.95)	−2.07 (−2.23, −1.90)	−68.19% (1,915,808)	−35.33% (1,136,515)	−59.28% (181,761)	−33.01% (105,384)
1st	Georgia: 1.10 (0.62, 1.59)	Georgia: 3.23 (1.77, 4.70)	Oman: −0.88 (−0.98, −0.77)	Cambodia: 1.99 (1.17, 2.81)	Qatar: 298.22% (105)	UAE: 393.91% (2627)	Qatar: 208.55% (6)	Qatar: 334.41% (13)
2nd	Cyprus: −0.01 (−0.12, 0.09)	Cambodia: 1.25 (0.53, 1.97)	Cambodia: −1.04 (−1.87, −0.20)	Georgia: 1.73 (0.84, 2.63)	UAE: 149.8% (482)	Qatar: 277.37% (115)	UAE: 133.34% (62)	UAE: 214.32% (86)
3rd	Singapore: −0.30 (−0.36, −0.23)	Mauritius: −0.60 (−0.99, −0.22)	Sri Lanka: −2.27 (−2.90, −1.63)	Oman: −0.04 (−0.06, −0.02)	Bahrain: 64.21% (48)	Cambodia: 101.86% (5664)	Bahrain: 97.73% (4)	Oman: 140.45% (10)
4th	Japan: −0.97 (−1.24, −0.70)	Sri Lanka: −0.92 (−1.47, −0.38)	Qatar: −2.63 (−2.78, −2.47)	Iraq: −0.05 (−0.27, 0.17)	Singapore: 59.13% (128)	Bahrain: 69.33% (148)	Oman: 61.17% (7)	Bahrain: 95.88% (5)
5th	Mauritius: −1.27 (−1.83, −0.71)	India: −0.93 (−1.37, −0.50)	Bahrain: −2.70 (−2.97, −2.43)	Uzbekistan: −0.14 (−0.15, −0.12)	Cyprus: 51.22% (31)	Oman: 66.31% (128)	Cambodia: −7.31% (1856)	Iraq: 93.59% (256)
*Death*								
Asia	−4.71 (−5.03, −4.38)	−3.41 (−3.54, −3.28)	−4.73 (−4.84, −4.63)	−3.13 (−3.25, −3.00)	−65.87% (31,792)	−30.21% (22,473)	−54.27% (3868)	−34.08% (1562)
1st	Georgia: 4.62 (2.71, 6.57)	Georgia: 4.15 (2.46, 5.87)	Cambodia: −0.59 (−1.47, 0.29)	Cambodia: 2.94 (1.98, 3.90)	UAE: 128.61% (6)	UAE: 447.05% (61)	UAE: 146.31% (1)	Oman: 282.22% (0)
2nd	Cambodia: −1.33 (−2.16, −0.49)	Cambodia: 1.83 (1.00, 2.67)	Oman: −1.89 (−2.42, −1.36)	Oman: 2.44 (2.07, 2.81)	Qatar: 28.12% (0)	Qatar: 278.41% (2)	Qatar: 120.79% (0)	UAE: 262.68% (1)
3rd	Pakistan: −3.57 (−3.74, −3.39)	Mauritius: −0.19 (−1.16, 0.79)	Sri Lanka: −2.28 (−2.94, −1.61)	Georgia: 2.37 (0.49, 4.29)	Bahrain: −7.11% (0)	Cambodia: 153.25% (137)	Bahrain: 67.41% (0)	Qatar: 254.27% (0)
4th	UAE: −4.33 (−4.50, −4.16)	Oman: −0.30 (−0.52, −0.09)	Nepal: −3.36 (−3.46, −3.26)	Iraq: 0.20 (−0.09, 0.49)	Georgia: −7.81% (1)	Oman: 128.62% (0)	Cambodia: 16.8% (47)	Cambodia: 157.4% (8)
5th	India: −4.46 (−5.04, −3.88)	Sri Lanka: −0.92 (−1.61, −0.22)	Qatar: −3.4 (−3.7, −3.1)	Sri Lanka: −0.05 (−0.75, 0.65)	Cambodia: −15.63% (118)	Bahrain: 79.04% (3)	Sri Lanka: −2.85% (2)	Iraq: 116.42% (3)
*Incidence*								
Asia	−0.06 (−0.08, −0.05)	−1.95 (−2.11, −1.08)	−0.91 (−1.18, −0.64)	−0.26 (−0.28, −0.24)	−0.24% (85,755,696)	−13.9% (53,022,856)	−3.5% (2,296,508)	7.51% (13,360,826)
1st	Thailand: 1.35 (0.99, 1.71)	Azerbaijan: −0.79 (−0.93, −0.64)	Armenia: 0.53 (0.45, 0.62)	Bhutan: 1.09 (0.84, 1.34)	Qatar: 323.67% (46,144)	UAE: 297.65% (49,152)	Qatar: 443.14% (2294)	Qatar: 437.76% (3116)
2nd	Japan: 0.86 (0.68, 1.03)	Afghanistan: −0.85 (−1.00, −0.71)	Mongolia: 0.41 (0.24, 0.57)	Yemen: 0.56 (0.36, 0.76)	Afghanistan: 246.45% (1,509,027)	Qatar: 282.41% (18,590)	UAE: 391.06% (7172)	UAE: 275.22% (9450)
3rd	South Korea: 0.37 (0.09, 0.65)	Japan: −0.85 (−0.95, −0.76)	Syria: 0.38 (0.06, 0.70)	Turkey: 0.46 (0.25, 0.67)	UAE: 145.54% (120,156)	Afghanistan: 152.9% (610,373)	Bahrain: 177.11% (1338)	Afghanistan: 236.32% (66,456)
4th	Nepal: 0.31 (0.27, 0.35)	Georgia: −0.99 (−1.14, −0.84)	Kazakhstan: 0.32 (0.28, 0.36)	Indonesia: 0.20 (0.11, 0.28)	Jordan: 117.81% (301,023)	Bahrain: 100.63% (12,765)	Jordan: 159.84% (7068)	Jordan: 163.64% (14,854)
5th	Bhutan: 0.28 (0.27, 0.30)	Laos: −1.00 (−1.18, −0.82)	Uzbekistan: 0.20 (0.07, 0.32)	Thailand: 0.12 (0.03, 0.20)	Palestine: 81.86% (157,536)	Yemen: 50.48% (562,645)	Afghanistan: 155.33% (48,471)	Yemen: 153.5% (33,506)

EAPC, estimated annual percentage change; DALY, disability adjusted life year.

## Data Availability

Not applicable.

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
