# Peer review of "Burden and Trends of Acute Viral Hepatitis in Asia from 1990 to 2019"

_viruses, 2022, doi:10.3390/v14061180_

Round 1
Reviewer 1 Report
In this article Liu et al are describing burden and trends of acute viral hepatitis in Asia in period of 1990-2019 using statistical data available from Global Burden of Diseases Study 2019 (GBD 2019) results.
Major comments:
While the article covers a long statistically covered period, a better introduction of methods used should be provided in the background.
Also while reading, it tends to be a bit confusing for the reader the mixed up data of DALY, ASMR, ASIR, ASDR for all hepatitis types explained. It would be much better if they would be described in separated sections for each, so the overall of each statistics data is easier to be understood by the reader, just like it was showed in the Figure 5.
Authors could also stress the difference in statistics between periods, in 2000s when the better treatments appeared comparing to the early 1990s.
Minor comments:
Reference labels should be better positioned, because they have a confusing effect on the following sentence, being at the end of the previous without the brackets.
The reference is missing for the lines 278-279-280.
The figures should be better labelled, not to be in the middle of one out of six graphs, but maybe on the side, that makes it visually easier.
Also, lower 4 graphs of the Figure 1 should be labelled properly , showing each color meaning, like in the upper one.
Author Response
In this article Liu et al are describing burden and trends of acute viral hepatitis in Asia in period of 1990-2019 using statistical data available from Global Burden of Diseases Study 2019 (GBD 2019) results.
Major comments:
While the article covers a long statistically covered period, a better introduction of methods used should be provided in the background.
Response: Thanks for the reviewer’s suggestion. We have added the introduction of methods used to estimate the disease burden of acute viral hepatitis in GBD 2019 as shown in line 91-101: “Briefly, all available data on causes of death, incidence or DALY data are standardized and pooled into a single database used to generate cause-specific estimates by age, sex, year, and geography; then multiple models, such as cause of death ensemble modelling, disease model-Bayesian meta-regression, comorbidity correction and so on were used to estimate comparable data of different diseases across the world [15]. In particular, acute hepatitis mortality was modelled by using vital registration, surveillance, and verbal autopsy data from the cause of death database; anti-HAV seroprevalence data, hepatitis B surface antigen seroprevalence data, anti-HCV seroprevalence data and anti-HEV sero-prevalence data from population-based studies and surveys was used to estimate sero-prevalence and seroincidence for further estimation of incidence and DALYs. [15]”
Also, while reading, it tends to be a bit confusing for the reader the mixed up data of DALY, ASMR, ASIR, ASDR for all hepatitis types explained. It would be much better if they would be described in separated sections for each, so the overall of each statistics data is easier to be understood by the reader, just like it was showed in the Figure 5.
Response: Thanks for the reviewer’s suggestion. We have described the data separately in section 3.1. However, in other sections, describing the data of ASMR, ASIR and ASDR together could help understanding the overall burden of acute viral hepatitis in each country or in each population group by sex and age. Nevertheless, we have added some words in the Results section to avoid being confusing.
Authors could also stress the difference in statistics between periods, in 2000s when the better treatments appeared comparing to the early 1990s.
Response: Thanks for the reviewer’s suggestion. We have added this statement in the discussion section as shown in line 291-294: “These decreasing trends could partly be explained by the rapid scientific and techno-logical advances in the last decades, which made it possible to control and even cure the infections, with a strong focus on prevention medicine through vaccination. [21]”
Minor comments:
Reference labels should be better positioned, because they have a confusing effect on the following sentence, being at the end of the previous without the brackets.
Response: Thanks for the reviewer’s suggestion. We have added the brackets for each reference label.
The reference is missing for the lines 278-279-280.
Response: Thanks for the reviewer’s suggestion. The reference of this sentence was “Chang M. L., Liaw Y. F. Overt acute hepatitis B is more severe in female patients. Hepatology 2017; 66(3): 995-6.” And we have cited it in this sentence.
The figures should be better labelled, not to be in the middle of one out of six graphs, but maybe on the side, that makes it visually easier.
Response: Thanks for the reviewer’s suggestion. We have changed the label position of Figure 2, 3 and 4.
Also, lower 4 graphs of the Figure 1 should be labelled properly, showing each color meaning, like in the upper one.
Response: Thanks for the reviewer’s suggestion. We have added the meaning of each color below the X-axis in Figure 1.
Reviewer 2 Report
Comments:
We know to achieve the goal of the 2030 Agenda for Sustainable Development, WHO made the Global Health Sector Strategy on Viral Hepatitis, 2016-2021, which aimed to eliminate viral hepatitis as a major public threat by 2030, and promoted synergies between viral hepatitis and other health issues and aligns the hepatitis response with other global health and development strategies, plans and targets. Identifying the burden of acute hepatitis is very important for the liver disease prevention and control strategies in different countries. This is a very useful story for readers and I suggest the Journal accepted this manuscript to publish after minor modified.
Minor modified:
- Abbreviation need to be show the real words in the first appearance.
- Different virus associated acute hepatitis has different epidemic history and outcome. It needs to be briefly described.
Author Response
We know to achieve the goal of the 2030 Agenda for Sustainable Development, WHO made the Global Health Sector Strategy on Viral Hepatitis, 2016-2021, which aimed to eliminate viral hepatitis as a major public threat by 2030, and promoted synergies between viral hepatitis and other health issues and aligns the hepatitis response with other global health and development strategies, plans and targets. Identifying the burden of acute hepatitis is very important for the liver disease prevention and control strategies in different countries. This is a very useful story for readers and I suggest the Journal accepted this manuscript to publish after minor modified.
Minor modified:
Abbreviation need to be show the real words in the first appearance.
Response: Thanks for the reviewer’s suggestion. We have added the meaning of abbreviations in the first appearance.
Different virus associated acute hepatitis has different epidemic history and outcome. It needs to be briefly described.
Response: Thanks for the reviewer’s suggestion. We have added the epidemic history and outcome as shown in line 42-45: “HAV and HEV are transmitted via fecal-oral route and rarely develop into a chronic infection. [1] HBV and HCV are transmitted via sexual contact and drug injections, and are the most common cause of liver cirrhosis, liver cancer and viral hepatitis-related deaths. [1]”